# The Job that Kills the Worker: Analysis of Two Case Reports on Work-Related Stress Deaths in the COVID-19 Era

**DOI:** 10.3390/ijerph20010884

**Published:** 2023-01-03

**Authors:** Maricla Marrone, Carlo Angeletti, Gerardo Cazzato, Gabriele Sebastiani, Luigi Buongiorno, Pierluigi Caricato, Fortunato Pititto, Eliano Cascardi, Alessandra Stellacci, Benedetta Pia De Luca

**Affiliations:** 1Section of Legal Medicine, Department of Interdisciplinary Medicine, University of Bari “Aldo Moro”, 70124 Bari, Italy; 2Section of Molecular Pathology, Department of Precision and Regenerative Medicine and Ionian Area (DiMePRe-J), School of Medicine, Aldo Moro University of Bari, 70100 Bari, Italy; 3Department of Medical Sciences, University of Turin, 10124 Torino, Italy; 4Pathology Unit, FPO-IRCCS Candiolo Cancer Institute, 10060 Candiolo, Italy

**Keywords:** work loads, COVID-19, death, physical illness, stress

## Abstract

The COVID-19 pandemic caused an increasing number of corporate layoffs and downsizing, as well as causing many employees to be absent due to illness, with inevitable consequences on the health of active workers both from a physical point of view, due to the need to make up for staff and organizational shortages, and from a mental point of view, due to the inevitable consequences related to the uncertainty of the social context. This context has certainly caused an increase in work-related stress, which is the pathological outcome of a process that affects workers who are subjected to excessive (emotional-relational or high or low or inadequate activity) or improper work loads. The purpose of this paper is to evaluate the main aspects of this issue, through the analysis proposed by two case reports, both of which occurred during the COVID-19 pandemic, in which occupational stress emerged as an etiological agent in the determinism of death.

## 1. Introduction

### 1.1. The Definition and Reasons for Work-Related Stress

Work-related stress is a condition that occurs when the demands in the workplace exceed the worker’s ability, physical or mental, to respond [1].

Work-related stress was defined by the European Agreement of October 8, 2004, implemented in Italy by an inter-federal agreement in 2008, which states that “*…work-related stress is a condition that may be accompanied by disturbances or dysfunctions of a physical, psychological or social nature and is a consequence of the fact that certain individuals do not feel able to respond to the demands or expectations placed on them…*” [2].

This condition can occur in any workplace regardless of the type of contract, size of company, field of work and in any worker, always taking into consideration the diversity that characterizes any person and the most varied ways of coping with strong pressures and increased workloads.

Job stress can be caused by a variety of reasons. These include:-Working conditions (exposures to physical and mental stressful situations: jobs involving long exposure to heat/cold, noise, toxic and/or hazardous substances, radiation, shifts that exceed the allowed hours, insufficient compensation, etc.);-Work organization (lack of work schedule planning, relationship with managers or colleagues, attitude between work and studies performed, etc.);-Communication in the work environment (prospects for career advancement, uncertainty of work shifts, etc.);-Subjective factors (insecurity, emotional pressures, dissatisfaction, etc.) [1,2].

Any worker is capable of responding to and withstanding greater workloads in a short interval of time, which, in this view, are considered favorable for the better development of the individual themself. Different, however, is the situation where such work pressures are prolonged for an excessive amount of time, and so much is configured as a stressful insult to the worker themself.

The latter can manifest work stress, in several ways:-Frequent mood swings;-Irritability;-Depression;-Migraine headaches;-Insomnia;-Gastro-intestinal disorders;-Difficulty with memory or concentration;-Increased use of alcohol, smoking and/or drugs, and psychotropic substances [1,2].

It seems clear, therefore, that increased worker dissatisfaction protracted over time will have inevitable repercussions on the worker’s work efficiency and mental and physical well-being.

It is mandatory, therefore, according to Framework Directive 89/391, that any employer *“…[ensure] the safety and health of workers in all work-related aspects...”* [3]; this includes occupational stress, as it constitutes a risk to the health and safety of the worker.

Any unexpected change in daily and work habits can affect the health of the worker; in this regard, the health emergency related to the spread of SARS-CoV2 completely subverted the previous organizational structure with the modification and introduction of new conditions regarding worker protection and safety, causing obvious repercussions in this area [4].

### 1.2. The Hazards of Work Stress in the COVID-19 Era

The effects of the pandemic in the work environment were the direct consequence of the reduction in individual freedom, the related measures to contain the contagion, and the adoptions of new and different ways of working, such as smartworking and video conferencing, etc. [5].

A study conducted by Restubog et al. shows that such a situation includes not only health care workers involved on the front lines during the pandemic, but all workers in different areas who have to endure more difficult working conditions due to limited availability of social or organizational supports, irregular working hours, and difficulties in finding adequate safety devices [6].

In addition, the economic recession associated with the COVID-19 pandemic caused more layoffs and corporate downsizing with inevitable consequences on workers’ health both from a physical point of view due to the need to make up for staff and organizational shortages and from a mental point of view due to the inevitable consequences related to the uncertain social environment [4].

The main causes of death caused by excessive stress related to a grueling work life are heart attack, stroke, cerebral hemorrhage, and heart failure; an aspect not to be underestimated is also the greater difficulty of access to care by workers who spend most of their time in the work environment [4].

### 1.3. The Epidemiology of Work Stress-Related Death in the COVID-19 Era

According to research by the International Labor Organization (ILO) and the World Health Organization (WHO), long working hours caused 745,000 deaths from ischemic heart disease and stroke in 2016, presenting a 29 percent increase since 2000 [4,7].

Specifically, from 2000 to 2016, the number of deaths from heart disease due to long working hours increased by 42 percent and the number of deaths from stroke increased by 19 percent.

The research shows that working more than 55 h per week is associated with a higher risk of ischemic heart disease and stroke, compared to working 35–40 h per week [6].

The results highlight that most of the victims (72%) were men between the ages of 60 and 74. Most affected were workers living in China, Japan, Australia, and Southeast Asia [7].

According to the 2021 Annual Report of Inail (Istituto Nazionale per l’Assicurazione contro gli Infortuni sul Lavoro, an Italian public agency that manages insurance against occupational accidents and diseases), there were 1361 fatal accident reports in 2021, a decrease of 19.2 percent compared to 2020 figures. Despite this decrease, however, the trend is far from reassuring in view of the fact that “traditional” fatal accident reports, excluding cases of SARS-CoV2 infection, increased by almost 10 percent compared to 2020 [4].

The purpose of this paper is to evaluate, through the analysis proposed by two case reports, both of which occurred during the COVID-19 pandemic, in which occupational stress emerged as an etiological agent in the determinism of death, the main aspects of this issue.

## 2. Materials and Methods

### 2.1. Case Report 1

In the pandemic era of COVID, a 57-year-old worker died after experiencing severe chest pain and presenting profuse sweating while climbing the stairs of the construction site where he worked.

From the testimony given by his wife, it was possible to deduce that he suffered from severe hypercholesterolemia and had experienced multiple episodes of moderate chest pain in the preceding days, which appeared whenever he exerted himself.

In addition, the same had complained about the continuous and exhausting shifts he was forced to work because many of his colleagues had tested positive for COVID-19 and thus, according to the regulations in force at the time, had been placed in home isolation.

The preventive occupational medicine tests to which the subject had been subjected had not, however, revealed any signs of pathology.

As he was on the morning of 20 April 2021, climbing the stairs of the construction site where he had been working now every day from 8 a.m. to 8 p.m. for the past month without interruption, he experienced severe chest pain and intense profuse sweating and collapsed to the ground. His coworkers alerted the police and rescue services, who ascertained his death at 09:20 (Figure 1A,B).

The cadaveric inspection performed by the medical examiners revealed the presence of scanty, pinkish-colored hypostases in the declivity and posterior regions of the trunk, as well as at the face and upper 1/3 of the chest (“cape-like”) [8]. Such “mantle-shaped” hypostases are one of the signs found in cases of cardiac death. Regarding this, it should be pointed out that the external cadaveric examination excluded the presence of recent major traumatic injuries [9,10].

Autopsy examination, however, showed a heart of normal shape but increased volume (transverse diameter 13 cm, longitudinal diameter 12.5 cm, antero-posterior diameter 4 cm), weighing 540 g. Macroscopic inspection revealed working spots (hypochromic areas) most represented at the septum and posterior wall of the left ventricle. The myocardium appeared of increased consistency, purplish-red coloring with evidence of diffuse dyschromic areas as from blood stasis and with increased left parietal thicknesses, in agreement with concentric hypertrophy (left ventricle—anterior wall 1 cm; lateral wall 1.1 cm; posterior wall 1.1 cm; septum 1 cm). In addition, serial transverse incisions on the course of the common trunk and left and right coronary arteries found diffuse sclerosis and the presence of plaques in the course of the common trunk and left coronary artery (Figure 2).

Therefore, it was possible to trace the death to acute heart failure on an arrhythmic basis, which certainly appeared as a result of work efforts, in agreement also with the circumstantial finding.

### 2.2. Case Report 2

The second case involves a 53-year-old man who was employed on night watchman shifts at an auto storage company. The same man alternated with two other guards in order to constantly watch over the premises both during the day and night. The same, however, had been forced to work daily and night shifts of 12 to 15 h for the past 20 days because one of the other wardens had contracted COVID-19 disease and had multi-organ complications that led him to the hospital. The man of this second case was found dead on 15 November 2021 at 8:00 a.m. on the couch of a room that the wardens could use during their work shifts (Figure 3A–D).

External examination of the body found the percolation of blood material from the oral cavity, which on opening appeared filled with it, and excluded the presence of recent traumatic injuries referable to the action of third parties.

Autopsy examination, on the other hand, revealed the presence of an aorta with diffusely increased consistency and with an intimal wall scattered with numerous plaques of various shapes and sizes, many of them calcific in consistency. In addition, a dissection of about 4 cm was found at the aortic arch and extra-pericardial descending tract, communicating with a false intramural lumen of about 5 to 6 cm in length (Figure 4A,B).

The heart, on the other hand, demonstrated thinned walls, a conspicuous increase in subepicardial adipose tissue, and the presence of working spots (hypopigmented areas) at the postero-medial wall of the left ventricle. In addition, increased myocardial thicknesses were found at the left sections (anterior wall vs. 1.5 cm; lateral wall vs. 1.5 cm; posterior wall vs. 1.5 cm; interventricular septum 1.2 cm), in agreement with concentric hypertrophy and diffusely sclerotic valves (Figure 5A–C).

Therefore, the cause of death was identified as hemorrhagic shock from rupture of a transverse dissecting aneurysm of the aortic arch (type A according to Stanford classification) and hypoxia secondary to hemorrhagic airway flooding.

## 3. Discussion

Sometimes, the worker, in order not to risk job loss, is willing to work continuously without daily breaks, vacations, or rest. At the same rate, the employer, by requiring its employees to work many extra hours, avoids new hires, new costs, and new constraints. So much causes, in some cases, a condition of considerable physical aggravation, in which work activity intensifies to an unsustainable pace [11].

The COVID-19 pandemic, as stated earlier, caused an increasing number of company layoffs and downsizing, as well as caused many employees to be absent due to illness, with inevitable consequences on the health of active workers both from a physical point of view, due to the need to make up for staff and organizational shortages, and from a mental point of view, due to the inevitable consequences related to the uncertain social environment.

Work-related stress is the pathological outcome of a process that affects workers who are subjected to excessive (emotional-relational or high or low or inadequate activity) or improper work loads. Maslach and Leiter in 2000 identified three behaviors that are established when one is in a stressful work environment [12]:-Deterioration of commitment to work;-Deterioration of emotions originally associated with work;-Inadequate fit between person and work.

Deaths occurring as a result of work-related stress conditions are mostly caused by heart attack, stroke, cerebral hemorrhage, and heart failure.

In fact, a meta-analysis published in the Lancet in October 2015 on a sample of more than 600,000 individuals [13] shows that an individual who works 55 or more hours per week, as opposed to those who work standard hours, has a 1.3-fold higher incidence risk of having a stroke and a higher incidence risk of going on to incident coronary artery disease.

Furthermore, throughout the scientific literature, and even more so in recent literature [14,15], stress and anxiety symptoms resulting from a stressful work environment are common denominators in patients with acute coronary artery disease, and may be associated with a substantial increase in cardiovascular morbidity and mortality [16].

In the two case reports cited in this article, which occurred during the COVID-19 pandemic, two deaths occurred, the first as a result of acute heart failure on an arrhythmic basis and the second as a result of hemorrhagic shock from the rupture of a transverse dissecting aneurysm of the aortic arch (type A according to Stanford classification) and hypoxia secondary to hemorrhagic airway flooding.

In fact, psychophysical stress can aggravate the state of sympathetic–adrenergic hyperactivity and thus promote hypertensive crisis and ventricular arrhythmias. Chronic stress induces, with the biohumoral changes it brings about, an acceleration of atherosclerotic processes at both coronary and systemic sites, while acute stress brings about functional changes, the epiphenomena of which are hypertensive crises and rhythm alterations with episodes of lethal cardiac tachyarrhythmia with or without myocardial infarction.

In both of these two case reports, it appears evident that the causes of death are with high probability secondary to work-related stress, in accordance with both the circumstantial data and the findings on external and autopsy examination.

There was no other anamnestic evidence to suggest a different etiological origin in either case. It would appear that death would not otherwise have occurred in the absence of work stress.

Stress in fact is among the modifiable risk factors of cardiovascular disease and can be taken as facilitating and aggravating coronary artery disease and ischemic heart disease; the same can be said of occupational stress, especially when it acts on individuals who, by their nature, can hardly bear work pressures, deadlines, or relationships with others [17].

Consequently, although the ways in which stress goes to increase cardiovascular morbidity and mortality have yet to be clarified, a correlation of heart disease with work-related stress appears evident in the two case reports cited above.

In particular, correcting one’s lifestyle assumes a critically important role in preventing heart disease. Indeed, stress is among the risk factors that affect an already present chronic disease, such as acute coronary artery disease and aneurysm, worsening its course and causing heart attack and aneurysm rupture, respectively. Such happened in the two case reports reported by the authors.

In Case Report 1, death occurred as a result of a heart attack; the subject had already complained of chest pain in the preceding days, but in order not to be absent from work to make up for the lack of personnel, had not gone to the emergency room, underestimating the pain he complained of. In addition, work stress goes to alter cardiovascular parameters and can act as a trigger for acute phenomena such as, for example, heart attack [18,19]

In Case Report 2, death occurred due to the rupture of an aneurysm. The subject had been forced at that time to make up for the lack of other work colleagues and to work night shifts every day lasting 12 to 15 h. Prolonged night work and excessive working hours have been scientifically shown to cause an increase in the hormone cortisol and the alteration of cardiovascular parameters, including blood pressure [20]. They may act as triggers to cause aneurysm rupture.

In both case reports, the causal link between the work activity and the death was found.

On the one hand, the absence of additional etiological noxae justifying the death in the manner and timing in which it occurred, and on the other hand, the abnormal workload, were sufficient elements to causally link work stress to the deaths.

In fact, the employer is liable for the worker’s death if they have not conformed their managerial–organizational power to the constraints imposed by Article 2087 of the Civil Code, in particular if they have subjected the worker to objectively burdensome workloads that exceed the limits of normal tolerability. In the two cases shown, the activity of the deceased workers had intensified to unsustainable limits incompatible with life.

## 4. Conclusions

In light of what has been argued, assuming that in both case reports, the cause of death is to be placed in clear correlation with occupational stress, it must be remembered that the employer has an obligation to know and assess all risks according to the regulations of Legislative Decree No. 81/2008. These risks include work-related stress risks, which are assessed through job content indicators, company indicators (accidents, sick leave, turnover, disciplinary proceedings, others) and work context indicators (career development, decision autonomy, others). All these indicators represent the basic parameters for preliminary risk assessment that every employer should know and control.

Work productivity, born as a life-enhancing tool for human beings, should not facilitate their death. Awareness-raising campaigns together with precise changes in work style are therefore of absolute importance, both in a pandemic and routine context, as the only way to prevent the biological determinism of stress-related illnesses.

## Figures and Tables

**Figure 1 ijerph-20-00884-f001:**
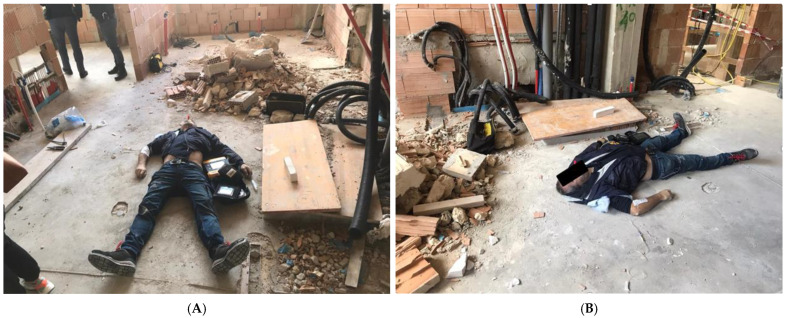
(**A**,**B**) Overview of the first patient reported. (**A**) frontal view; (**B**) lateral view.

**Figure 2 ijerph-20-00884-f002:**
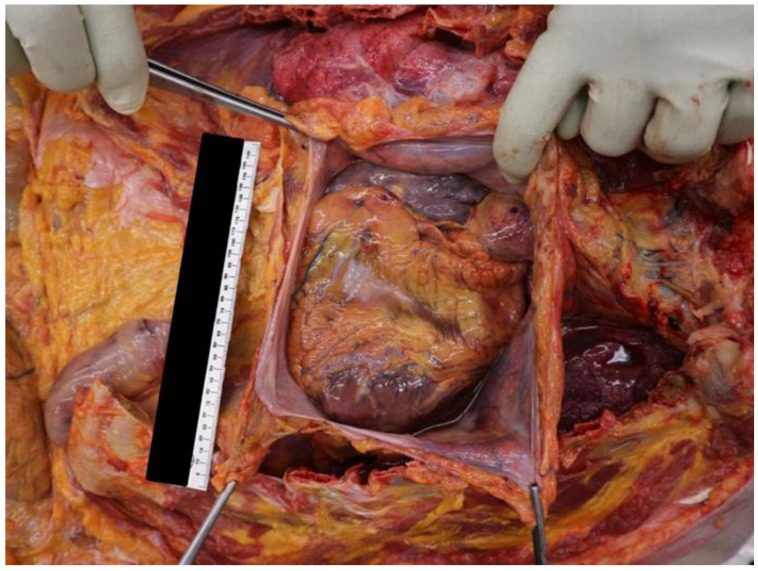
Macroscopical details of the heart and pericardium of the patient.

**Figure 3 ijerph-20-00884-f003:**
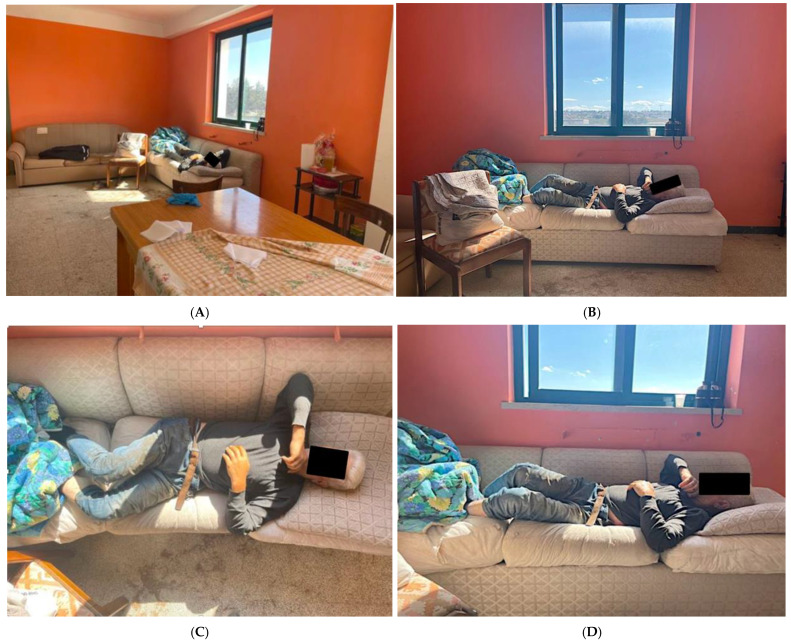
(**A**–**D**) Different views of the of the body found without life. (**A**) panoramic view; (**B**) lateral view; (**C**) view from top; (**D**) lateral view.

**Figure 4 ijerph-20-00884-f004:**
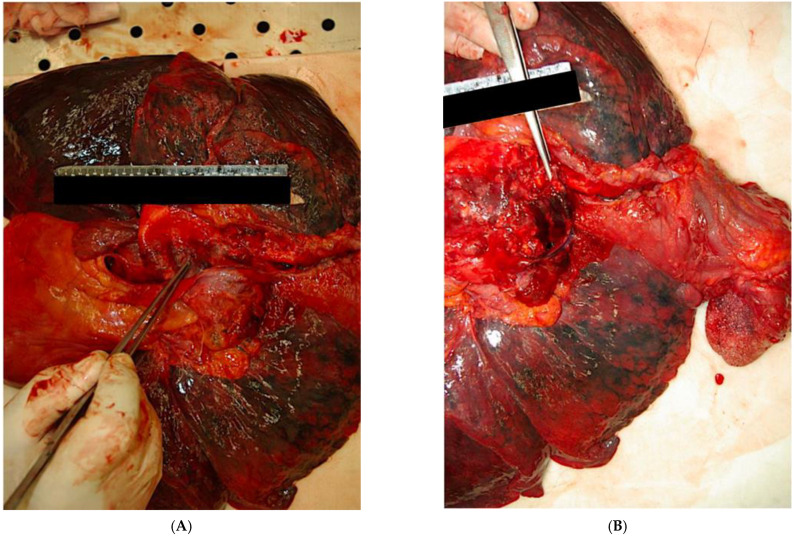
(**A**,**B)** Macroscopic details of the heart and lungs. Macroscopic features of dissection recorded to autoptic examination.(**A**) lateral vies; (**B**)view from top.

**Figure 5 ijerph-20-00884-f005:**
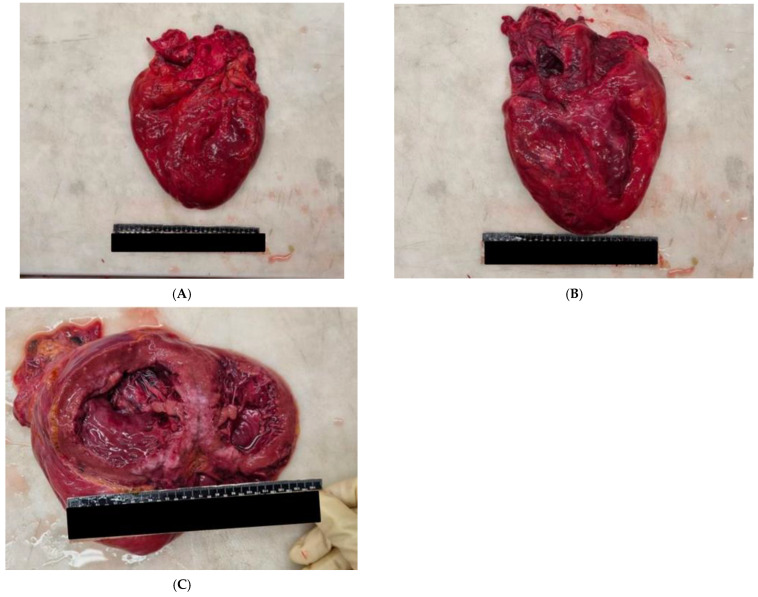
(**A**–**C**) Autoptic features of the heart: note the presence of increased myocardial thicknesses at the left sections consistent with concentric hypertrophy and also diffusely sclerotic valves.(**A**,**B**) view from top; (**C**) coronal section.

## Data Availability

Not applicable.

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
