# Peer review of "The Job that Kills the Worker: Analysis of Two Case Reports on Work-Related Stress Deaths in the COVID-19 Era"

_ijerph, 2023, doi:10.3390/ijerph20010884_

Round 1
Reviewer 1 Report
This manuscript discussed two case reports to analyze work-related stress-induced deaths in the Covid-19 era, concluding that occupational stress emerged as an etiological agent in the determinism of death. This may provide a warning for individuals' physical and mental health in the context of the COVID-19 epidemic. However, the quality of this manuscript can be improved by rectifying the following flaws.
Major points:
1. In the “Introduction” part, there are too many paragraphs to make the author's main topic unclear. For the consideration of prominent emphasis and clear structure, please revise the paragraphs into three:
i. The definition and reasons for work-related stress;
ii. The hazards of work stress in the COVID-19 era;
iii. The epidemiology of work stress-related death in the COVID-19 era.
2. In the “Discussion” part, most of the content have been mentioned in the “Introduction” part. Also, there are too many paragraphs in this part. Please simplify the discussion to make it more concise.
3. In addition to the summary of the two cases, the discussion part should conduct a preliminary exploration of the pathological mechanism underlying these two cases and the implications for the future.
4. The “Conclusion” section lacks a medical conclusion.
5. Please confirm the “Author Contributions”.
6. Please refer to the format requirements for references from “International Journal of Environmental Research and Public Health” and revise the reference format.
Minor points:
1. Lines 2-3: The first letter of a content word should be capitalized in the title.
2. Lines 39-46: Lack of references.
3. Lines 52-61: Lack of references.
4. Lines 87-89: Lack of references.
5. Lines 91-93: Lack of references.
6. Lines 94-96: Lack of references.
7. Lines 97-99: Lack of references.
8. Lines 103-108: Lack of references.
9. Lines 115-128: These three paragraphs can be merged as one.
10. Lines 129,161,162,163,174,183,184: All the figures mentioned were lack of figure legends or titles. If they belong to a group of figures, please add “A” “B”, like: Figure 1A, and Figure 1B.
Author Response
v
eviewer n’1: In the “Introduction” part, there are too many paragraphs to make the author's main topic unclear. For the consideration of prominent emphasis and clear structure, please revise the paragraphs into three:
- The definition and reasons for work-related stress;
- The hazards of work stress in the COVID-19 era;
iii. The epidemiology of work stress-related death in the COVID-19 era.
Answer n’1: Dear Reviewer n’1, thank you very much for your useful advices to improve the quality of our manuscript. Done, thanks a lot.
Reviewer n’1: In the “Discussion” part, most of the content have been mentioned in the “Introduction” part. Also, there are too many paragraphs in this part. Please simplify the discussion to make it more concise.
Answer n’2: Thanks again. Done.
Reviewer n’1: In addition to the summary of the two cases, the discussion part should conduct a preliminary exploration of the pathological mechanism underlying these two cases and the implications for the future.
Answer n’3: Dear Reviewer n’1, thank you. We emphasised the pathological mechanisms (sympathetic iper-activation) that are at the base of death-stress-related.
Reviewer n’1: The “Conclusion” section lacks a medical conclusion.
Answer n’4: Dear Reviewer n’1, we have added 'medical' statements to our conclusions. Thank you.
Reviewer n’1: Please confirm the “Author Contributions”.
- Please refer to the format requirements for references from “International Journal of Environmental Research and Public Health” and revise the reference format.
Answer n’5: Done, thank you.
Reviewer n’1: . Lines 2-3: The first letter of a content word should be capitalized in the title.
- Lines 39-46: Lack of references.
- Lines 52-61: Lack of references.
- Lines 87-89: Lack of references.
- Lines 91-93: Lack of references.
- Lines 94-96: Lack of references.
- Lines 97-99: Lack of references.
- Lines 103-108: Lack of references.
- Lines 115-128: These three paragraphs can be merged as one.
- Lines 129,161,162,163,174,183,184: All the figures mentioned were lack of figure legends or titles. If they belong to a group of figures, please add “A” “B”, like: Figure 1A, and Figure 1B.
Answer n’6: Done.
Reviewer 2 Report
Dear authors, comments are given in the file below.

Author Response
Dear Reviewer No. 2, thank you very much for your compliments and your question. We have added this information, emphasising the fact that the visits from occupational medicine had not revealed any pathological signs
Round 2
Reviewer 1 Report
The manuscript can be accepted.